# Innovative Leaf Area Detection Models for Orchard Tree Thick Canopy Based on LiDAR Point Cloud Data

Chenchen Gu [1,2,3], Chunjiang Zhao [1,2,3], Wei Zou [1,2,3], Shuo Yang [1,2,3], Hanjie Dou [1,2,3] and Changyuan Zhai [1,2,3,*]

1  Intelligent Equipment Research Center, Beijing Academy of Agriculture and Forestry Sciences, Beijing 100097, China
2  Research Center of Information Technology, Beijing Academy of Agriculture and Forestry Sciences, Beijing 100097, China
3  National Engineering Research Center for Information Technology in Agriculture, Beijing 100097, China
*  Correspondence: zhaicy@nercita.org.cn; Tel.: +86-10-51503886

**Abstract:** Orchard spraying can effectively control pests and diseases. Over-spraying commonly results in excessive pesticide residues on agricultural products and environmental pollution. To avoid these problems, variable spraying technology uses target canopy detection to evaluate the leaf area in a canopy and adjust the application rate accordingly. In this study, a mobile LiDAR detection platform was set up to automatically measure point cloud data for a thick canopy in an apple orchard. A test platform was built, and manual measurements of the canopy leaf area were taken. Then, polynomial regression, back propagation (BP) neural network regression, and partial least squares regression (PLSR) algorithms were used to study the relationship between the orchard tree canopy point clouds and leaf areas. The BP neural network algorithm (86.1% and 73.6% accuracies for the test and verification data, respectively) and the PLSR algorithm (78.46% and 60.3%, respectively) performed better than the Fourier function of the polynomial regression (59.73% accuracy). The leaf area model obtained using PLSR was intuitive and simple, while the BP neural network algorithm was more accurate and could meet the requirements for high-precision variable spraying.

**Keywords:** leaf area detection model; thick canopy; target-oriented spray; BP neural network; partial least squares regression; LiDAR

## 1. Introduction

Orchard spraying can effectively control diseases and insect pests and makes it possible to harvest 66% to 90% of the fruit, which is very important in orchard operation management [1]. However, over-spraying is common and results in the waste of pesticides, environmental pollution, and excessive residues on agricultural products [2–5]. A lack of information about the target geometry and density of orchard trees is the main reason for excessive spraying. Variable spraying, according to information on the canopy position, volume, and outline of orchard trees, improves the accuracy of spraying [6–10]. Biomass detection is also very important in the process of variable spraying [11–13]. Canopy biomass can be detected by sensors, such as ultrasonic sensors, light detection and ranging (LiDAR), RGB cameras, and the Microsoft Kinect sensor [14–16]. RGB cameras and the Microsoft Kinect sensor can be easily influenced by sunlight [16], and the process of image processing is complex. At present, ultrasonic and LiDAR sensors are mainly used to study biomass in the canopy. Ultrasonication strength can reflect the density of the canopy. The time-domain energy method was used to study the canopy density by Li et al. (2017), whose work was carried out in the laboratory using artificial canopies of known density with multiple layers of leaves [17]. Nan et al. (2019) built a cylindrical support with multiple layers of leaves and proposed a target leaf area density calculation method based on an ultrasonic sensor echo signal [18]. The leaf area density model of the simulation canopy was obtained,

the canopy leaf area density of an Osmanthus tree was measured using the echo signals of the ultrasonic sensor, and the model feasibility was validated with the measurement. Palleja et al. (2017) applied point quadrat analysis (PQA) to study the canopy density of orchard trees through ultrasonic detection [19]. The PQA method measures the contact points between the branches, leaves, and fruit of orchard trees and the probe rod, and does not need to count individual leaves. A linear relationship model between the ultrasonic echo energy and sample square data was obtained. This method, which is nondestructive and has a fast measurement speed can measure the biomass of different regions and provide references for ultrasonic measurements. As ultrasonic sensors are easily affected by the beam angle and external environment in the process of target detection, their suitability for practical applications is poor.

The leaf area is the key component of tree biomass. During the growth processes of orchard tree canopies in a single year, there are both small changes in canopy volume and large changes in leaf area, and the demand for pesticide application changes greatly with large changes in the canopy leaf area. Leaf area is the basis for spray deposition density calculation in the NY/T 992-2006 standard [20]. The spray volume is required to achieve a spray deposition density of 25 droplets/cm$^2$ for a low-volume spray and 70 droplets/cm$^2$ for typical fungicides. Research on canopy leaf area detection models can provide a useful basis for determining the spray volume. Such models also serve as important references for controlling wind during air-assisted variable-rate spraying. A LiDAR sensor has good stability and is rarely influenced by weather conditions, and point cloud data can be calculated quickly. Using a small banyan tree in the laboratory and a modern dwarf espaliered apple tree, Sanz et al. (2011) developed linear relationship models between canopy leaf area and LiDAR point cloud data [21]. The canopy of the orchard trees studied by Sanz et al.(2011) was a wall-type canopy with uniform thickness, and the trees originated from dwarf rootstock. The applicability of the leaf area model to orchard tree canopies with large thicknesses and uneven canopies needs to be further verified. Zhang et al. (2017) studied the relationship between the number of LiDAR point clouds and simulated tree leaf area in the laboratory [22]. Simulated canopies with different numbers of branches and leaves were experimentally studied. The authors concluded that the relative error was different with different density of branches and leaves. However, the experiment was carried out indoors with a simulated tree and there was only a single data source. Thus, the feasibility of this leaf area model in practical applications needs to be verified. Mahmud et al. (2021) developed a canopy density detection model for two varieties of apple orchards [23]. Linear regression models between automatic point counts and manual leaf counts were established. The coefficient of determination ($R^2$) of the model at site one was lower than that at site two; the main cause was the lower leaf foliage density of the trees, with less overlap at site two. Berk et al. (2020) studied 20 apple trees, of 3 different varieties, in orchards [24]. Canopy point cloud data and canopy leaf area were obtained by scanning the orchard tree canopy with LiDAR and manually picking leaves. The relationships among the point cloud number, canopy volume, and canopy leaf area were studied using regression analysis. A canopy primary leaf area detection model was obtained as an equation. Due to the overlap of leaves in the canopy, the correlation between the LiDAR point cloud data and leaf area was poor. It was necessary to further integrate different sensors or intelligent algorithms to improve the calculation accuracy of the canopy leaf area of orchard trees [17,19,25].

The complexity of the establishment of the leaf area relation model, the feasibility of application, and the hardware requirements of the algorithm are the factors that need to be considered in practical applications. Models based on this relationship are important for orchard-oriented variable spraying technology. Most of the previous studies used conventional regression analysis to study the relationship between the LiDAR point cloud number and canopy leaf area, in cases with thin canopies, dense leaf distributions, and simple statistics. The relationship between the LiDAR point cloud number and canopy leaf area was easy to obtain, and the established leaf area models were simple and revealed a

strong correlation. However, the applicability of these studies to orchard trees with thicker canopies and sparse leaf densities needs to be further verified. When using LiDAR to detect thick canopies, the following problems were encountered. The laser beam measured only the outermost part of the orchard tree canopy and could not detect the interior of the orchard tree; the penetration was poor. Due to the influence of the internal branches of the orchard tree canopy, there was a discrepancy between the LiDAR point cloud number and the leaf area data. Thus, it is more challenging to develop a leaf area detection model with a large canopy thickness using LiDAR.

Since the canopy of orchard trees in flat areas of apple orchards are generally thick, which is the traditional planting pattern, the existing canopy leaf area detection model has poor applicability. In this manuscript, leaf area detection models for thick orchard tree canopies, based on the LiDAR point cloud number, were studied. The conventional polynomial regression, back propagation (BP) neural network regression, and partial least squares regression (PLSR) algorithms were used. The purpose of this paper is to provide a reliable basis for accurately calculating the quantity of precision variable spraying and to promote the further development of precision variable spraying technology.

## 2. Materials and Methods

### 2.1. Test Platform and Method for Measuring Leaf Area

2.1.1. Orchard Detection Object

The test was conducted at the Xiaotangshan National Precision Agriculture Research and Demonstration Base in Changping District, Beijing (Longitude: 160°26.695′; Latitude: 40°10.806′). A test tree is shown in Figure 1. The height of the tree was 2.3 m, the lower edge of the canopy was 0.8 m from the ground, and the crown was 1.5 m high and 2.5 m wide. The test tree was a Fuji apple tree, the canopy density was approximately average for the orchard, and the tree was 5 years old. The experiment was conducted in July 2020. According to the definition of the pome fruit growth stages of mono- and dicotyledonous plants (Meier, 2001), the canopy of the orchard tree was in principal growth stage 8: maturity of fruit and seed.

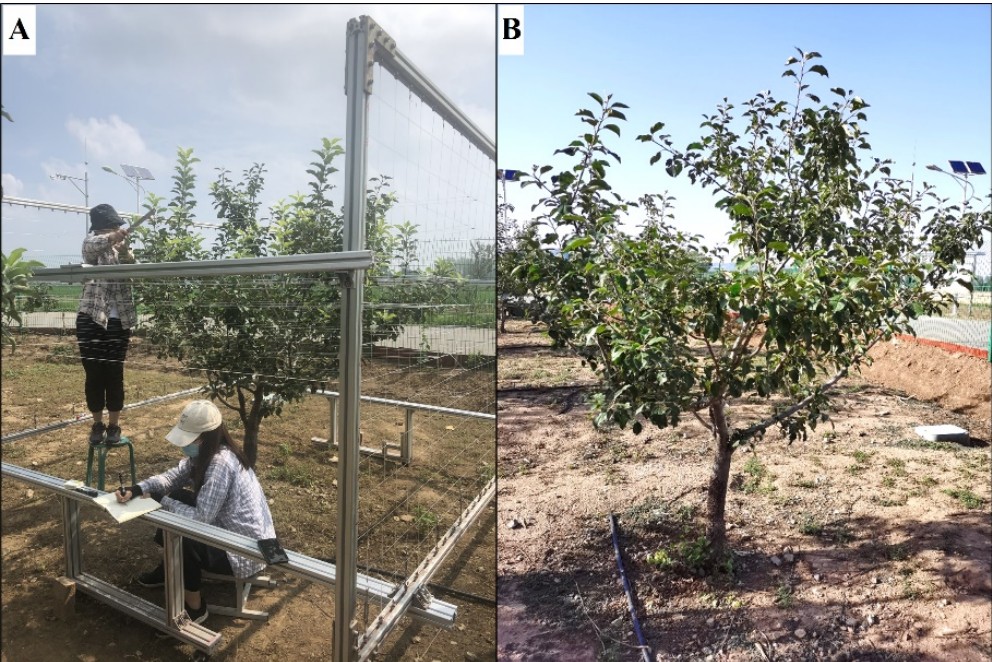

**Figure 1.** Manual measurement process of canopy leaf area (**A**) and the test orchard tree (**B**).

2.1.2. Experimental Platform for the Three-Dimensional Manual Measurement of Leaf Area

A three-dimensional experimental platform for canopy leaf area measurement was designed (Figure 2). The platform was composed of a grid frame, support, and grid line. The grid dimensions of the grid frame were 1.5 m high and 4 m wide, the grid mesh was 0.1 m, the height of the bracket was 2.07 m, and the distance between the front and rear grid frames was 3.5 m. The platform was equipped with a tensioning device around the grid, which could tighten the grid longitudinally and horizontally, to ensure the horizontal and longitudinal tensioning of the grid lines. The grid lines were made of transparent nylon line with a diameter of 1 mm, to divide different areas of the orchard tree canopy and minimize the influence of the grid line on LiDAR point cloud data acquisition.

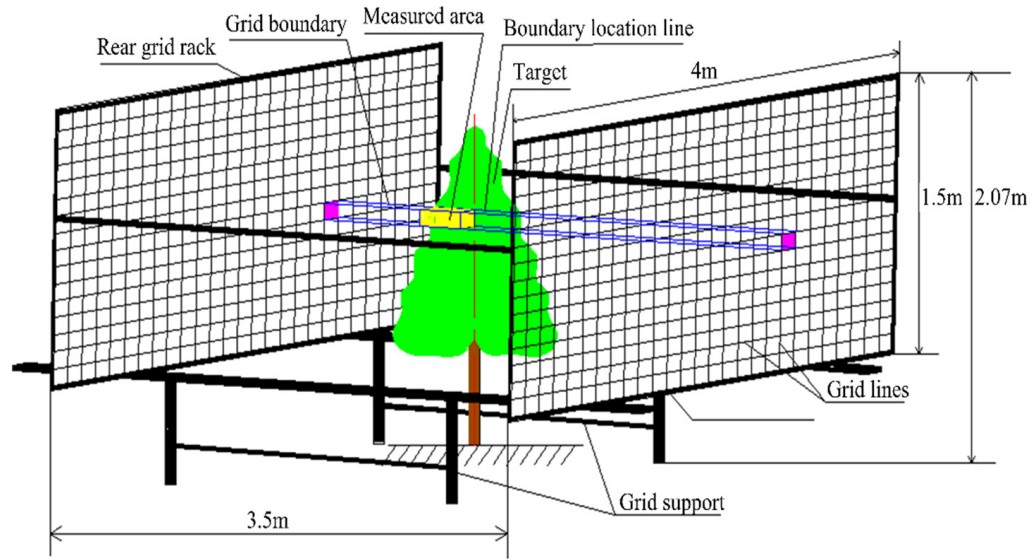

**Figure 2.** Schematic diagram of the three-dimensional experimental platform structure and measurement method for canopy leaf area.

2.1.3. Statistical Method for Determining the Leaf Area in the Grid

The methods of canopy leaf area measurement include destructive measurements, nondestructive in situ measurements, and sampling and statistical measurements. A destructive measurement involves picking all the leaves on an orchard tree, measuring the area of each leaf with a measuring instrument, and calculating the total. This method is accurate, but it is destructive to orchard trees, and the test process is not repeatable. Nondestructive in situ measurements use a measuring instrument to determine the area of each leaf in the canopy. This method is suitable when the number of canopy leaves is small and easy to measure, but it is not suitable for the measurement of canopy leaf areas with dense leaves.

In this experiment, the method of sampling and statistical measurement was used to calculate the canopy leaf area. A total of 219 leaves of different sizes and shapes were randomly selected in the orchard (Figure 3). The leaves were divided into three size grades: large, medium, and small. Ten leaves were selected from each grade. The leaf area was measured with a leaf area meter (YMJ-G, factory number: FK20200731001; manufacturer: Shandong Fangke Instrument Co., Ltd., Weifang, China). The leaf area of each stage was measured and weighted to average. For the leaves at the grid boundaries, two-thirds of the leaves were recorded in the grid, whereas the remaining leaves were counted in the next grid. The number of leaves were 76, 55, and 88 in the large, medium, and small size grades, respectively. The leaf area of each grade was multiplied by the number of leaves, and the leaf areas of the three grades were added to obtain the total leaf area. The area ranges of large, medium, and small level leaves were 17.12–29.83 cm$^2$, 11.01–18.87 cm$^2$, and 3.22–12.92 cm$^2$, respectively. This leaf area was compared with the total value measured

by the instrument, and the relative error was 1.8%. This result proves that the proposed method is feasible and appropriate for this experimental study.

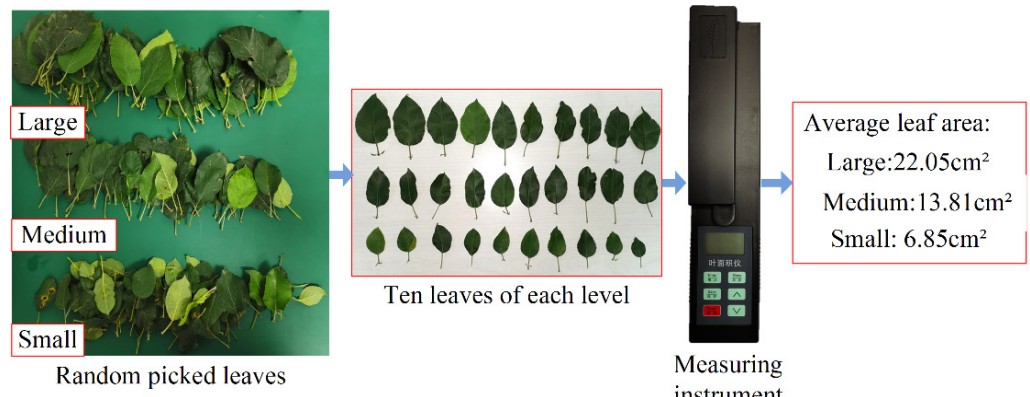

**Figure 3.** Leaf area measurement using statistical analysis.

The real leaf area of the apple trees was calculated in three steps. First, the leaves in the statistical area are divided into three grades: large, medium, and small. Second, count the number of leaves in each grade. In the last step, the calculated average leaf area of each grade was multiplied by the number of leaves in that grade. The leaf areas calculated for the three grades were summed, and the total leaf area was obtained. The specific implementation steps were as follows: the boundaries of the measurement areas were first detected (Figure 2), and then the numbers of leaves in the different size grades in the measurement area were manually counted.

### 2.2. Test Platform and Method for LiDAR Detection

#### 2.2.1. LiDAR Mobile Detection Experimental Platform

The LiDAR mobile detection experimental platform (Figure 4) was used to scan the canopy of orchard trees, to obtain canopy point cloud data. A Sick LMS111–10100 2D LiDAR sensor (SICK AG, Baden-Württemberg, Germany) was used in this system. The centerline of the LiDAR detection mobile platform was 2 m ($L_1$) from the centerline of the orchard tree canopy. LiDAR detected the driving distance of the mobile test platform (from the start position of the sign board to the end position) at 5 m. The detection speed was 0~1.2 m/s, and the sensors was 1.5 m from the ground. V indicates the moving speed of the LiDAR during the detection process. The row spacing of orchard trees was 4 m ($L_2$). During the test, a level meter was used to adjust the platform to a horizontal position. The detection speed of the LiDAR on the slide rail was set to automatically collect the point cloud data of the orchard tree canopy. The coordinate system of the LiDAR laser scanner and the details of the test platform can be found in Gu et al. (2021) [5]. In the spraying process, spray was applied along each row of the trees, one-half of each tree was detected and sprayed, and then the other half of the tree was sprayed when the sprayer moved to the next row.

#### 2.2.2. Data Processing

We used LiDAR to obtain point cloud data of the orchard trees by moving the slider along the slide rail of the mobile detection platform. According to the canopy gridding method, the obtained point clouds were gridded using MATLAB, and the number of LiDAR points in different grid regions was counted. The number of LiDAR points corresponded to the manually measured leaf area in the corresponding grid region, and the original data were obtained. Due to the influence of orchard tree branches and other factors in the orchard tree canopy, there were unreasonable data points in the relationship between the leaf area value and point cloud number. The residual method was used to remove the unreasonable data in the original data. Polynomial regression, BP neural network regression, and the

PLSR algorithm were used to analyze the experimental data and develop the leaf area detection model. During the experiment, the orchard trees were divided into 240 regions, according to a 0.1 m × 0.1 m grid, and 240 groups of leaf area data and LiDAR point cloud data of the corresponding regions were obtained.

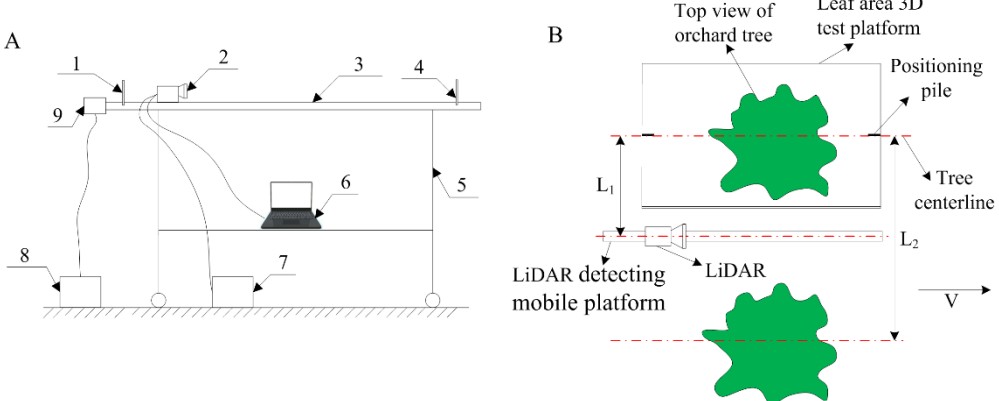

**Figure 4.** LiDAR detection mobile platform and schematic diagram: (**A**) schematic diagram, including the starting position of the sign board (1), LiDAR (2), linear rail guide (3), end position of the sign board (4), mobile platform (5), computer (6), power supply (7), stepper motor controller (8), and stepper motor (9); (**B**), LiDAR detection mobile platform operation, top view.

### 2.3. Removal of Outliers from the LiDAR Point Cloud Array and Raw Leaf Area Data

The original data set was preprocessed to remove the outliers in the data and obtain the potential value of the test data [26]. GB/T 4883-2008 (2008) prescribes different processing methods and rules for outlier data processing, which provided a reference basis for the removal of outlier values in this experiment [27], e.g., by means of extreme value removal, standardization, neutralization, residuals, and so on. To remove data, it is first necessary to determine the normal distribution of the data; otherwise, the process does not directly exclude the outliers and risks removing correct data, which affects the accuracy of the data model [28]. In this study, the residual method was used to remove the outliers from the original experimental data. There are four basic steps for the outlier removal: calculate the parameters of the outlier removal model and the data residuals, judge the outlier, and remove. The formula used to calculate the parameters of the outlier removal model is:

$$[b, bint, r, stats] = regress(y, x1) \tag{1}$$

where $b$ is the estimated value of the coefficient of multivariate linear regression, bint is the lower and upper limits of the confidence boundary of the estimated coefficient, $r$ stands for the residual value, and *stats* is a vector with four parameters in the model statistics.

$b$ is a $p \times 1$ vector, and the dimension $p$ is the same as the number of variables. The first column of bint contains the lower limit of the confidence boundary of each coefficient estimate, and the second column contains the upper limit of the confidence boundary, which is also a $p \times 1$ vector. r stands for the residual value, which is an $n \times 1$ vector. $n$ is the number of observations in the independent variable. The 95% confidence interval of each data point used to diagnose outliers is rint, which is an $n \times 2$ matrix. If the rint (i,:) interval of observation i does not contain a zero value, then the corresponding residual is greater than the expected residual of the new observation of $100 \times (1 - alpha)$%, indicating that there is an outlier [29]. Alpha is a significant and positive scalar, stats is a vector with four parameters in the model statistics, and the four parameters are $R^2$, F statistics, significance $p$ value, and the error variance estimation value.

The residual presents a normal distribution with zero mean and has different variances for different values of predicted variables. The residual is the regression function, and the residual value is divided by the estimated value of the standard deviation independent

of its value, so that the residual can be compared on a scale. The rcoplot (r, rint) function is used to draw the residual diagram of the data fitted by the regression () function for regression analysis, the outliers in the data group are identified, and the residual diagram of the data is made. The process of removing outliers requires several iterations, and each iteration is carried out based on the removal of the previous outlier. Outliers are removed one by one, until all outliers are removed. To determine the number of iterations for the reasonable removal of outliers, it is necessary to analyze the relationship between the number of outliers removed and the number of iterations.

### 2.4. Model Analysis of the Relationship between LiDAR and LA

The relationship between LiDAR point cloud data and LA was studied. The iterative test data were reordered randomly and divided into a test set and a verification set at a ratio of 7:3. The test set was used for modelling, and the verification data set was used to validate the regression model. Polynomial regression, BP neural network regression, and PLSR algorithms were used to fit and model the data after outliers were removed, and a comparative analysis was carried out to determine the efficacy of the outlier removal and the improved accuracy of the relational model.

### 2.4.1. Polynomial Regression Algorithm

The MATLAB cftool toolbox was used to analyze the data, and five functions were used to study the detection model of LiDAR point cloud data and canopy leaf area. The polynomial function, exponential function, Fourier function, Gaussian function, and sum of sine function were used for equation fitting calculations. Considering the influence of the complexity of the algorithm on the real-time performance of the variable spray system, the primary function was selected for fitting.

### 2.4.2. BP Neural Network Regression Algorithm

A BP network is a multilayer feedforward neural network that is characterized by signal forward transmission and error BP and can train the weights of nonlinear differentiable functions. In the forward pass, the input leaf area values in each size grade ($x_1, x_2 \ldots x_n$) are processed layer by layer, from the input layer to the output layer, the corresponding LiDAR points values are graded ($y_1, y_2 \ldots y_n$), and the state of the neurons in each layer only affects the state of the neurons in the lower layer. That is, one vector is read from the input vector at a time, and each feature dimension is an input node. The value of each dimension is assigned to the hidden layer, according to the weight, and the hidden layer is multiplied by the weight and then transmitted to the output node through the activation function. If the desired output is not obtained in the output layer, it is transferred in a backward pass, and the weight between the input layer and the hidden layer is corrected according to the prediction error (the difference between the output layer value and the real value). In this study, the input layer is the matrix corresponding to the leaf area and LiDAR point cloud data, and the output layer is the leaf area prediction matrix.

A BP network was used to calculate the leaf area detection model for the data with outliers removed. The mapminmax function was used to normalize the data (Formula (2)), to eliminate the difference in the order of magnitude between different dimensions of data. If the row data were the same, for example, in the case of $x_{max} = x_{min}$, let $y = y_{min}$, the two values $y_{max}$ and $y_{min}$ default to −1 to 1. At the same time, the data do not change.

$$y = \frac{(y_{max} - y_{min})(x - x_{min})}{x_{max} - x_{min}} + y_{min} \tag{2}$$

According to the actual data format, the BP network was set to a 1-1-1 grid structure, that is, 1 input layer, 1 output layer, and 1 hidden layer. The basic principle of determining the number of hidden layer nodes is to select the most compact structure possible, while maintaining the required accuracy; that is, the number of hidden layer nodes is as small as

possible, and the number of nodes $p$ must be less than N − 1 (N is the number of training samples). The empirical formula based on the number of nodes is expressed as follows:

$$p = sprt(m + n) + a \tag{3}$$

where $m$ is the number of nodes in the input layer, $n$ is the number of nodes in the output layer, and $a$ is an adjustment constant between 1 and 10.

According to the calculation, the number of nodes should be in the range of 2~11, and it was set to 3 after the adjustment test. Considering the time of the BP neural network training process and the requirements for computer resources, the number of training iterations was set to 1000, the minimum error of the training target was 0.01, and the learning rate was set to 0.01. The trainlm function (LM: Levenberg–Marquardt algorithm) was selected in the newff function to create the BP neural network, which is the default function for learning and training the BP neural grid and has the advantage of the fastest training speed for medium-scale grids. The sample data test set and verification set were divided randomly, and the mean square error was used to measure the network performance.

### 2.4.3. PLSR Algorithm

PLSR combines the characteristics of principal component analysis, canonical correlation analysis, and linear regression analysis. This approach can provide a reasonable regression model, and can complement the results of principal component analysis and correlation analysis. PLSR studies the influence of the sample population on the predicted value, based on fully considering the comprehensive effect of the individual factors on the predicted value. Its regression rate is faster than that of the general multiple regression method [30,31], with the advantages of simple calculation, high prediction accuracy, and convenient qualitative interpretation [32].

## 3. Results

### 3.1. Distribution Map Corresponding to the Original Data

The corresponding relationship between the LiDAR point cloud data and leaf area in the original data is shown in Figure 5. Most of the data shown in the scatter plot are concentrated, although a small number of data points are scattered. The main reason for the presence of scattered points is that the number of actual LiDAR points was relatively high, because there were more branches and stems in the individual measurement areas, and the leaves were at relatively lower heights. The points corresponding to the leaf area were located close to the x-coordinate axis. Due to the large density of leaves in each measurement area and the substantial amount of overlap, the amount of LiDAR point cloud data was small. The points corresponding to the actual leaf area were far from the *x*-axis. Due to these phenomena, the original data set could not reflect the overall distribution of the leaf area.

### 3.2. Data Normality Analysis

The next step was to determine whether the data conformed to the normal distribution, before removing outliers from the test data. MATLAB software was used for this process. The norm plot function was used to draw the normal distribution probability diagram of the original data (Figure 6), and the sample data in the figure are represented by "+". The probability diagram shows the relationship between the cumulative frequency distribution of the sample and the cumulative probability distribution of the theoretical normal distribution. Figure 6 shows that the graphic display is approximately linear, and the data obey a normal distribution.

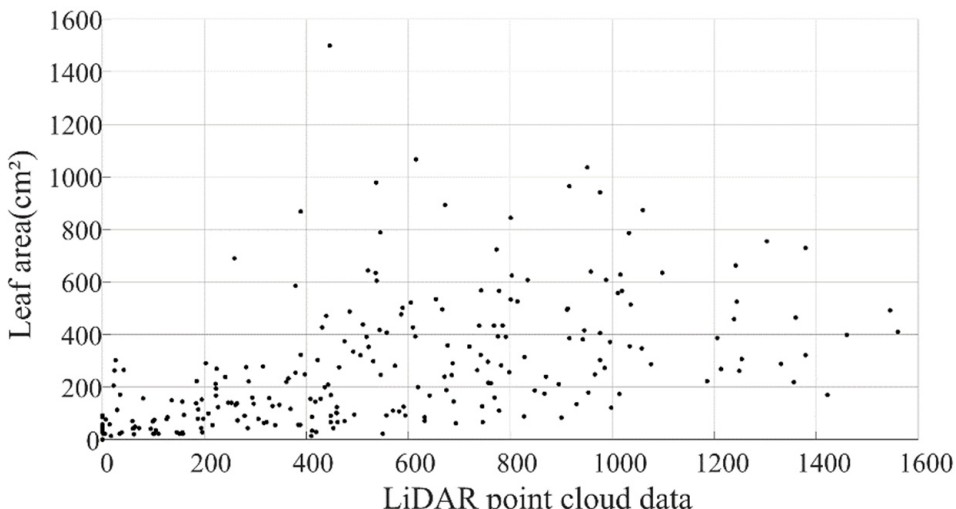

**Figure 5.** Original LiDAR point cloud data and leaf areas.

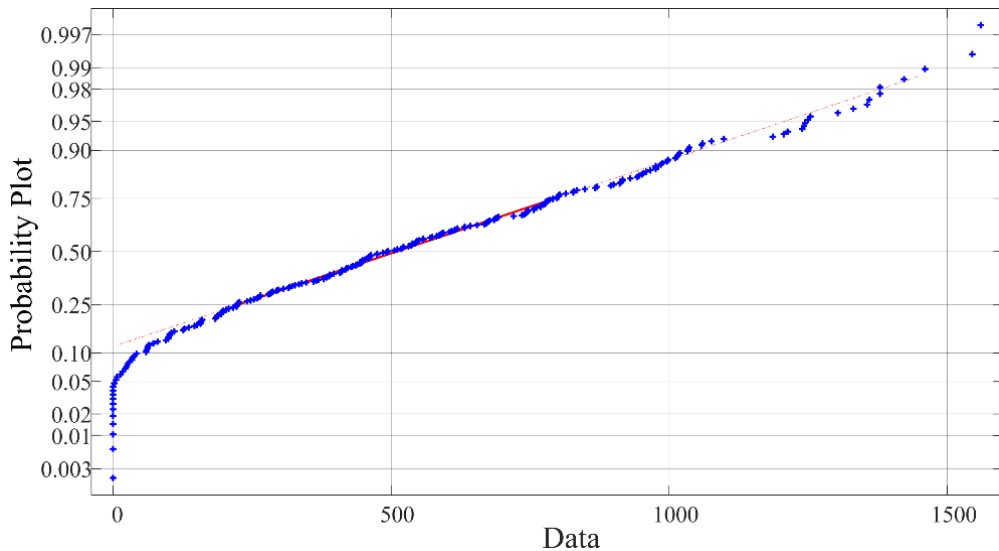

**Figure 6.** Probability diagram of the normal distribution of data.

*3.3. Outlier Removal Result*

The residual graph after the second iteration and the removal of outliers and residual values is shown in Figure 7. The red lines in Figure 7A,C indicate the presence of outliers, and the number of red lines indicates the number of outliers. In Figure 7D, red, blue, and black indicate the outliers removed in the first iteration, the outlier removed in the second iteration, and the remaining data, respectively. Each iteration was performed based on the previous removal of outliers. In the next iteration, residual analysis was performed on the data after the iteration, and the data whose residual values exceeded the standard were removed.

The result of data iteration is shown in Figure 8, which was completed after a total of 17 iterations. After the 17th iteration, only one point was removed from the data for five consecutive iterations. Thus, the number of iterations was complete after the 17th iteration, and all outliers had been removed. The $R^2$ value in the comparative statistical analysis model Stats gradually increases with the increase in the number of iterations (1–17); the value increased from 26% after the first iteration to 74.4% after 17 iterations, an increase of 48.4%. After the 18th iteration, the $R^2$ value of the data decreased and was 73.95%. The $p$ value of $8.9 \times 10^{-39}$ in Stats was less than 0.05, and the data had a significant regression level. There was a significant regression relationship between the LiDAR point cloud data

and the leaf area. Through the above analysis, the outliers were completely removed after 17 iterations of the data. This shows that the outliers removed by the residual method were reasonable, and that this data processing method can be applied to this research.

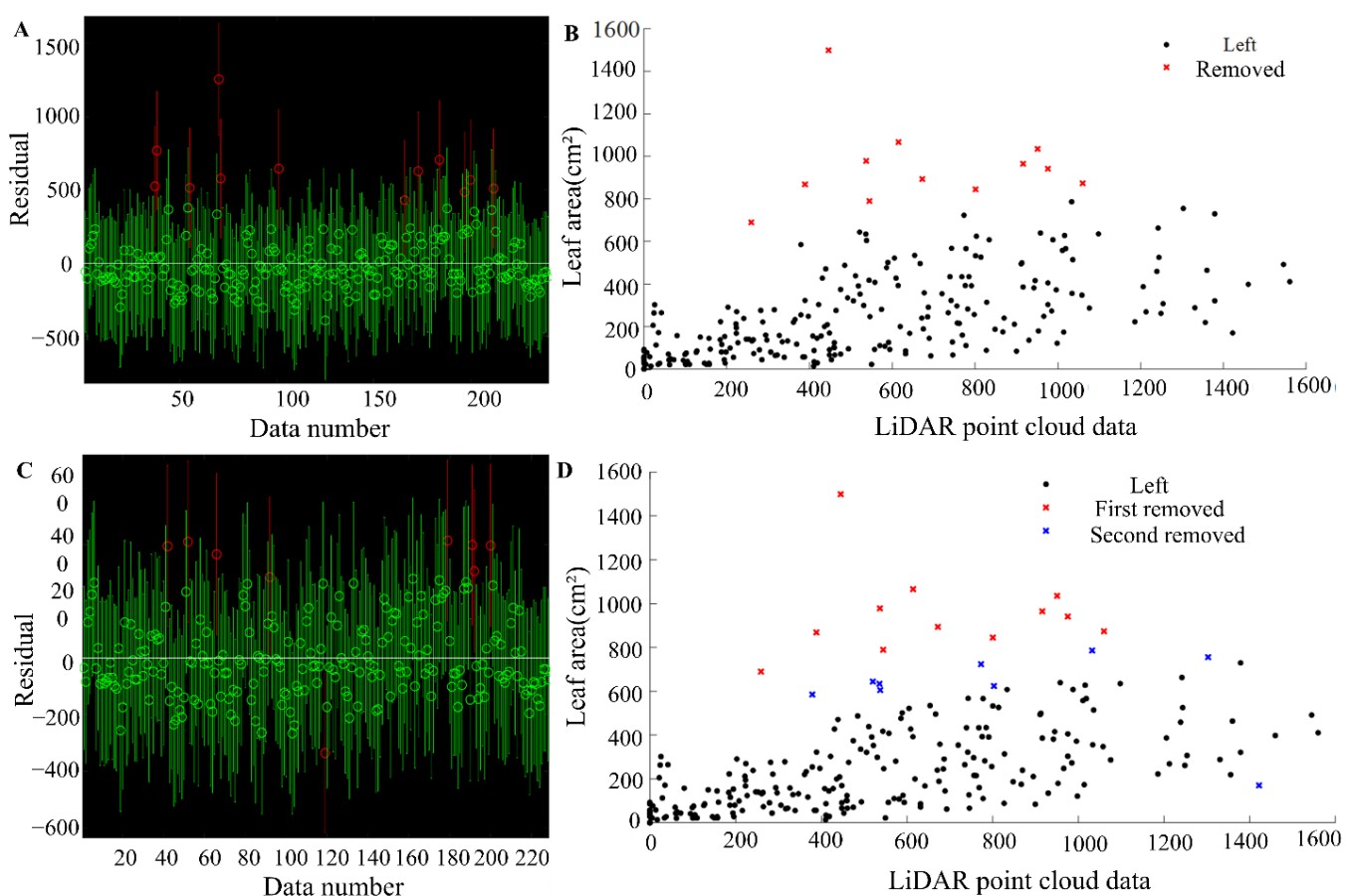

**Figure 7.** Comparison of residual maps and data after removal of outliers after one and two iterations. Residual graph from the first iteration (**A**) and the second iteration (**C**). (**B,D**) Comparison of the removal of outliers and residual values after one iteration and two iterations, respectively.

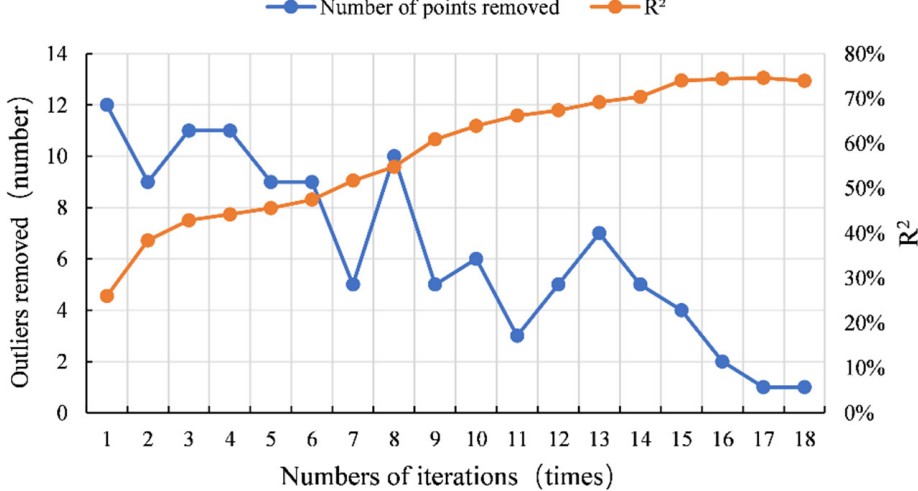

**Figure 8.** Relation between iteration times and number of outliers removed.

The outliers removed after the 17th iteration and the data are shown in Figure 9.

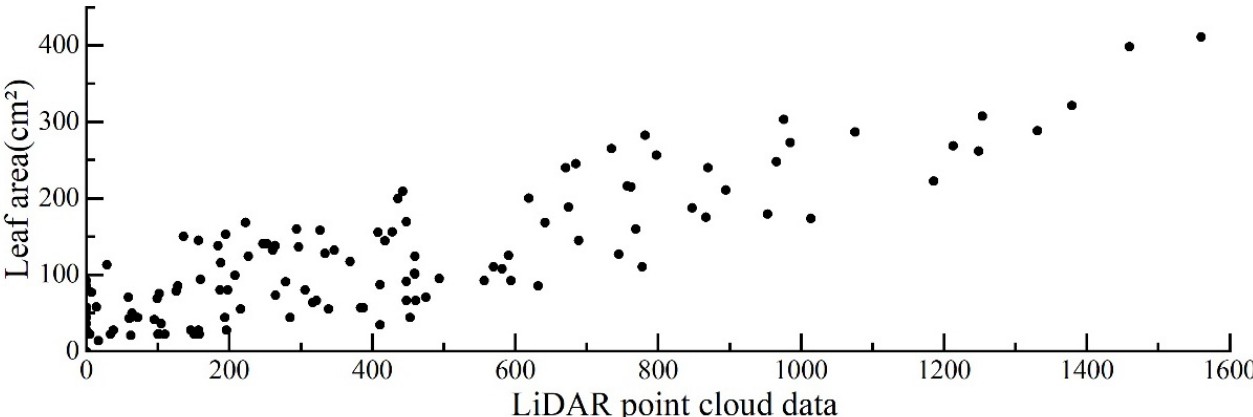

**Figure 9.** The residual values after seventeen iterations.

*3.4. Polynomial Regression Analysis Results*

The result of the equation diagrams obtained with the five functions is shown in Figure 10. This figure shows that there is a one-to-one correspondence between the dependent variable and the independent variable. As the value of the LiDAR point cloud data increases, the corresponding leaf area increases, which intuitively reflects the true relationship within the data and conforms to the real-world situation. The obtained graphs illustrate the importance of removing outliers in a reasonable and effective manner. The $R^2$ value of the Fourier function and Gaussian function was greater than the $R^2$ of the other three functional fitting equations during the whole iteration process, which is consistent with the data distribution. The equation of the Fourier function was the best, and the maximum $R^2$ value was 79.4%. The smaller the RMSE value, the more concentrated the data [33,34]. Comparing the RMSE of the equation reveals that the Fourier function had the smallest RMSE, which indicates that the equation was optimal. This result is consistent with the results of the above mentioned $R^2$ value comparison. Analyzing the $R^2$ and RMSE of the equation reveals that the Fourier function is better than the other equations.

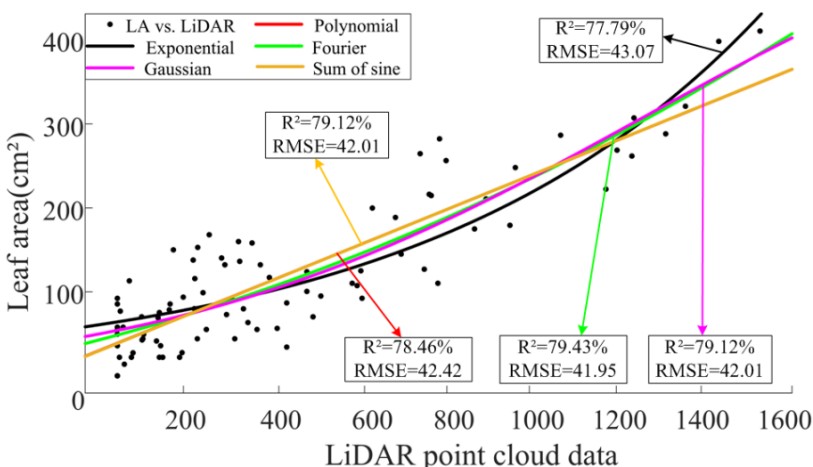

**Figure 10.** Comparison of fitting equations of test data.

In summary, through comparative analysis, the Fourier function showed an advantage and can be used as a model for calculating the relationship between LiDAR and canopy leaf area data. The equation is:

$$y = 3.912 \times 10^9 - 3.192 \times 10^9 \times \cos\left(1.639 \times 10^{-7}x\right) + 8.123 \times 10^5 \times \sin\left(1.639 \times 10^{-7}x\right) \quad (4)$$

where $y$ is the leaf area value (cm$^2$) and $x$ is the LiDAR point cloud data (points).

### 3.5. BP Neural Network Regression Model for the Calculation of the Relationship between LiDAR and Leaf Area Data

The training process of the BP neural network regression model is shown in Figure 11. In the calculation process, mu is the error accuracy, which is used to add another modulation to the weight of the neural network, to avoid falling into a local minimum during the training of the BP neural network. Its range is 0–1. Epoch is the training times. The progress bar shows the actual training times. Time is the time used in this training. Performance refers to the maximum mean square error (mse). The progress bar shows the current mean square error, and the right shows the set mean square error. If the current mean square error is less than the set value, training is stopped. Gradient is the current gradient value displayed in the progress bar, and the set gradient value is displayed on the right side. If the current gradient value reaches the set value, training is stopped. Generalization ability check is a validation check. The output error obtained after the validation of the sample data is input is obtained. Generally, the number of steps of the validation set is set to 6. If the error does not decrease, but instead rises for six consecutive training iterations, the training error of the training set is no longer reduced, and then the training is forcibly ended, to prevent overtraining.

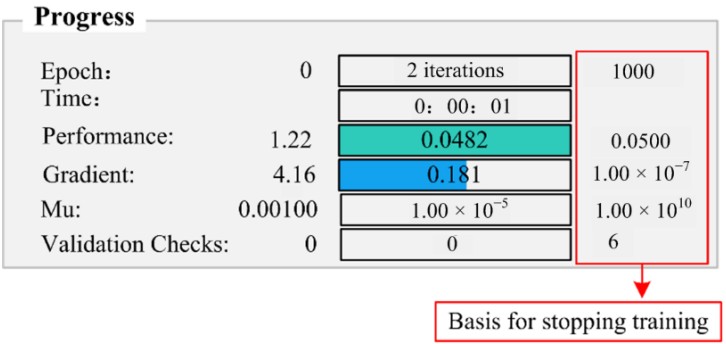

**Figure 11.** BP net training process.

The calculation result of the obtained BP neural network regression model prediction set is shown in Figure 12. The red and blue lines are the predicted and measured values, respectively. The curve shows that the gap between the two sets of data is relatively small, and the trends are relatively consistent. The predictive ability of the BP neural network leaf area regression model on the prediction set was 79.36%, and the data fitting effect was good. Figure 13 shows the prediction correlation analysis of the overall data obtained by the BP neural network model.

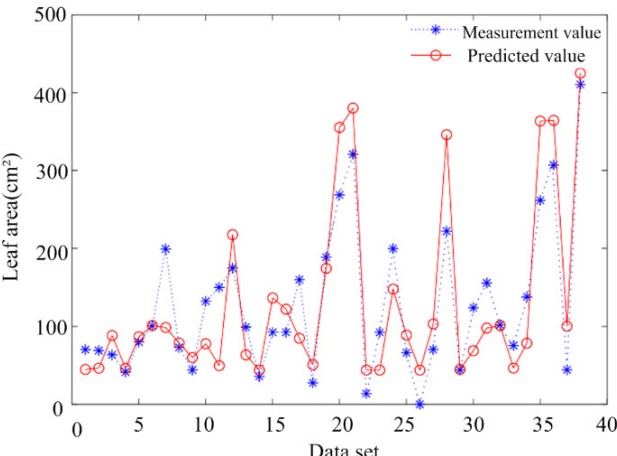

**Figure 12.** Comparison between the real leaf areas and predicted values in the data prediction set.

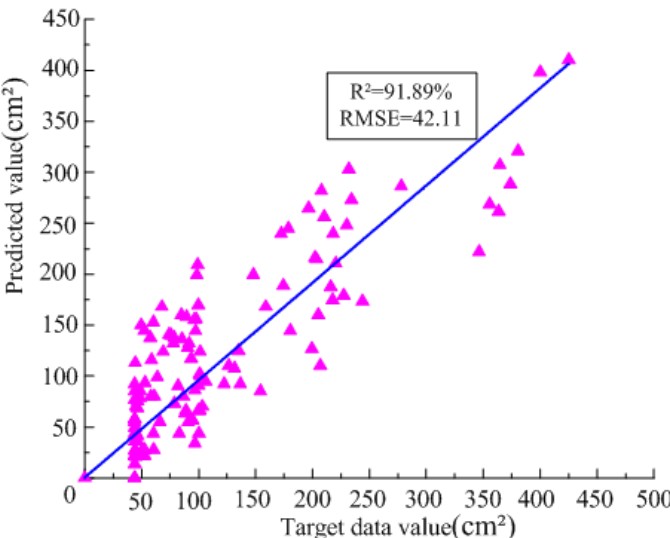

**Figure 13.** Relationship between the real and predicted values of the BP neural network test set.

The predicted value and real value are obtained through model calculations. The relationship model is shown in Formula (5):

$$Y = 0.956T \tag{5}$$

where $Y$ is the predicted value of the data, and $T$ is the target data value.

The $R^2$ of the target data and predicted values of BP neural network test set was 91.89%, and the RMSE was 42.11. The $R^2$ of the leaf area detection to the model was 89.87%, the accuracy of the model was higher than that of the Fourier function obtained by polynomial regression, and the accuracy of the model was improved by 10.47%. According to the above analysis, the BP neural network regression model had a good predictive ability and a strong ability to explain the data. This shows that the accuracy of the leaf area regression model could be effectively improved by using the BP neural network regression model, and the leaf area detection model obtained was better and could be applied to the calculation of the leaf area for a variable spray.

*3.6. PLSR Algorithm Calculation of the Leaf Area Detection Model*

Figure 14 shows the results of data analysis using the PLSR algorithm. The red line in the figure is the predicted value of the leaf area detection model based on the measurement set data, and the blue line is the measurement value used for modelling the leaf area detection model. The figure shows that the predicted value of the model on the measurement set data was relatively close to the measured value, and the change trends were similar. The $R^2$ of the obtained model was 78.46%. Compared with the polynomial regression model, the $R^2$ value for the PLSR model was greater than the exponential equation, equal to the polynomial and sum of sine function equations, and smaller than the Gaussian and Fourier functions. The $R^2$ values of the leaf area detection model obtained by polynomial regression and the leaf area detection model obtained by PLSR were both smaller than those of the BP neural network.

Figure 15 is a curve comparing the obtained leaf area detection model to the prediction set data and the true value. There is a gap in the curve changes between the two groups. The fitting accuracy of the obtained model to the prediction set data is 60.3%.

The $R^2$ of the target data and predicted values of the PLSR test set was 91.08%, and the RMSE was 42.42. The accuracy of the model was higher than that of the polynomial regression but lower than BP neural network regression model.

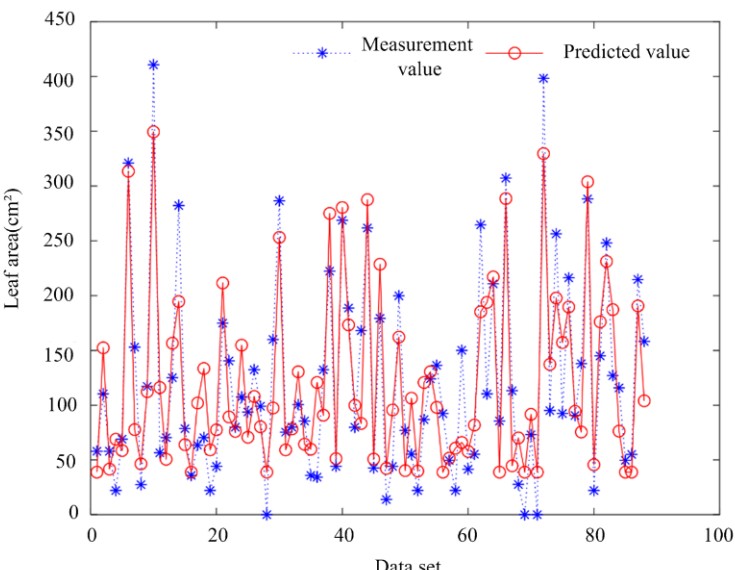

**Figure 14.** Test set data fitting diagram.

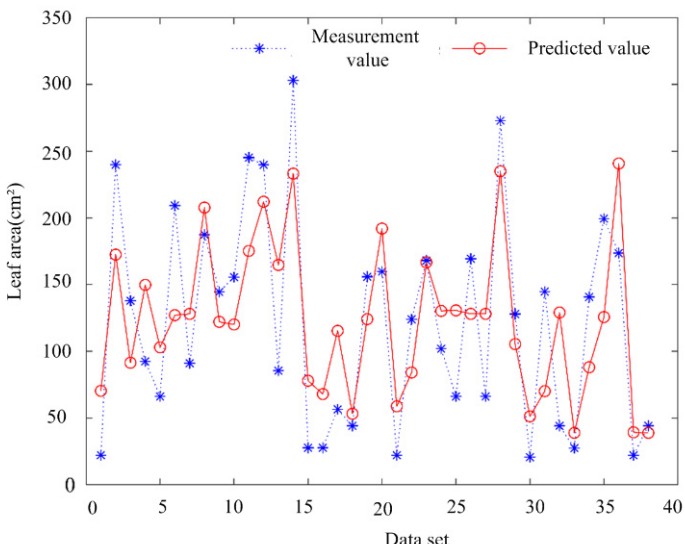

**Figure 15.** Validation set data prediction results.

The regression equation obtained by the PLSR algorithm is

$$y = 0.199x + 38.841 \tag{6}$$

where $y$ is the leaf area value (cm$^2$) and x is the LiDAR point cloud data (points).

The predicted value and real value are shown in Figure 16. The relationship is Formula (7):

$$Y = 0.910T \tag{7}$$

where $Y$ is the predicted value of the data, and $T$ is the target data value.

In summary, the fitting equations of the polynomial regression, BP neural network, and PLSR algorithm for different iteration times and that were used to remove outliers were compared, and leaf area detection models were obtained. Comparing the three regression algorithms shows that the R$^2$, RMSE, and relative error of the verification set of the five functional equations obtained by polynomial regression were obtained, and the Fourier function was the best. The R$^2$ value of the measurement set in the function model of the BP neural network and the R$^2$ value of the model validation set were both higher. The R$^2$

values of the leaf area detection model obtained using polynomial regression and the PLSR algorithm were lower than those of the regression model of the BP neural network.

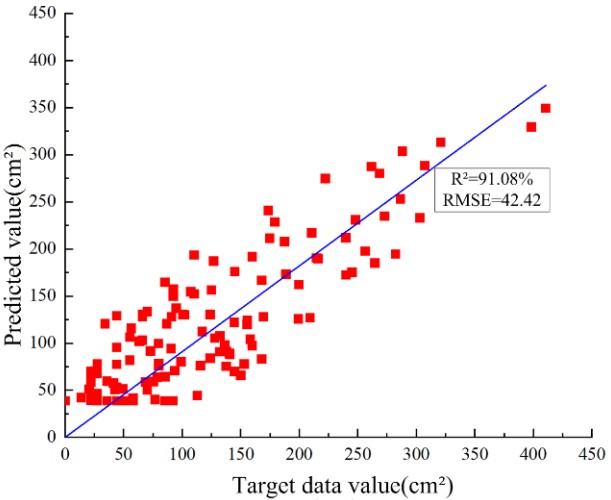

**Figure 16.** Relationship between the real and predicted values of PLSR test set.

## 4. Discussion

This study focuses on the problem of thick orchard tree canopies and the difficulty in establishing a relationship model between the LiDAR point cloud and canopy leaf area data. Since the laser beam does not penetrate the orchard tree canopy, due to the influence of the branches and trunks, and because the distribution of leaves in different positions in the orchard tree canopy differs, the individual plots of the obtained LiDAR point cloud numbers have unusual specificities, since individual anomalies obscure the relationship for the entire data set. If these abnormal points cannot be reasonably found and removed, they will bias the data and lead to faulty conclusions about the studied correlations. The residual method was used to remove anomalous data and noise from the measurements and eliminate their effects during the establishment of the model. After 17 iterations, 126 sets of experimental data were obtained, the $R^2$ value in the model statistics Stats was 74.4%, an increase of 48.4%. Compared with the 36 data sets of Sanz et al. (2011) [21], the measurement data repeated three times for 84 areas by Zhang et al. (2017) [22], the 20 data sets of Mahmud et al. (2021) [23], and the 20 data sets of Berk et al. (2020) [24], our data, with 126 sets, exhibited strong reliability and could reflect the internal relationship between the number of LiDAR point clouds and the canopy leaf area.

Classical polynomial regression, the BP neural network regression algorithm, and PLSR, which have obvious regression advantages, were used to study the relationship between LiDAR point cloud data and canopy leaf area and obtain leaf area detection models. The $R^2$ value of the model obtained using the PLSR algorithm was 78.46%, which is 0.97% lower than the Fourier function equation $R^2$ value of 79.43%, i.e., a difference of less than 1%. The validation set was used to verify the fitting effect of the model. The $R^2$ of the Fourier function fitting to the prediction set data was 59.73%, and the $R^2$ of the PLSR fitting the prediction set data was 60.3%. The predictive ability of PLSR for the prediction set data was better than that of the Fourier function. The model obtained using the PLSR algorithm was a linear equation, which is simpler than the equation of the leaf area detection model obtained by the Fourier function. The purpose of studying the apple tree canopy leaf area relationship model is to support variable spraying. The higher the accuracy of the model obtained, the simpler the model, and the more conducive it is to the realization and implementation of system control. When models have similar accuracies, then the simplest model is the first choice. The above comparative analysis revealed that the leaf area detection model obtained using the PLSR algorithm is better than the Fourier function and has stronger applicability. The accuracy of the leaf area detection model obtained using the BP neural network was 86.1%, and the fitting ability of the obtained model to the

validation set data was 73.6%, which is the best among the three algorithms. In the process of data regression using polynomial regression and the PLSR algorithm, the regression equation obtained was relatively intuitive, whereas the BP neural network cannot directly calculate the regression equation. The process of obtaining the functional equation through polynomial regression depends on the equation type selected, and whether the selection of the equation type is appropriate has a considerable impact on the accuracy of the obtained model. The calculation process of PLSR and the polynomial regression model is highly dependent on the mathematical knowledge of the operator. For functional equations with insufficient regression accuracy, stepwise regression is required. In the process of BP neural network training, to obtain a better model and accuracy, reasonable parameters need to be set. Multiple regression training was used to approximate the objective function. The BP neural network cannot intuitively reflect the relationship between an input and output, and can construct a model of the relationship between an input and output only by drawing points. A BP neural network can be used for complex model regression, especially when the relationship between variables in the model cannot be determined. The above analysis revealed that the BP neural network regression algorithm was better than the PLSR algorithm overall; the model fitting accuracy was 7.46% higher than that of the PLSR algorithm, and the prediction on the verification set was 13.3% higher. As the leaf area detection model obtained using the BP neural network algorithm is not intuitive, PLSR has the advantages of providing an intuitive and simple model. There was a 7.46% difference in the accuracy of the two models, which would cause a certain error during the spraying process; however, the overall analysis showed that this approach can still save a large quantity of pesticide in most cases and improve the accuracy of spraying. In the variable spraying selection process, the leaf area detection model based on the BP neural network or PLSR can be selected, according to the required spraying accuracy and acceptable spraying error range.

Sanz et al. (2011) used conventional regression analysis to study the relationship between LiDAR point cloud data and canopy leaf area in dwarf rootstock orchards [21]. The leaves were densely distributed on the trunk, the canopy layer was thin, and the fitting accuracy of the leaf area model was high. Zhang et al. (2017) [22] studied irregular artificial trees indoors. The canopy was thin, the leaves were densely distributed in the canopy, and the spaces between the leaves were small, which could better correlate with the LiDAR point cloud data. The test process used data from three repeated scans, and the data were highly repeatable. In contrast, the research in the present study focused on orchard trees in standard apple orchards. The canopies of orchard trees are irregular, with thick canopies and uneven distributions of canopy branches and leaves. The data are representative and adaptable. In this study, a statistical analysis method was used to obtain the canopy leaf area, which is better than the destructive measurement research used in previous studies [24,35], and this approach allows rapid and accurate measurement.

This study developed models of the number of LiDAR point clouds and leaf area data in the thick canopy of an orchard. Through the removal of outliers, the influences of poor laser beam penetration and canopy stems on the leaf area detection model were avoided. Considering the poor applicability of existing regression models and the urgent need for advanced algorithms, an intelligent algorithm BP neural network and PLSR regression algorithm were introduced. Compared with the results of polynomial regression, the leaf area detection models obtained using the PLSR and BP neural network regression algorithms were more advantageous. The leaf area detection model obtained in this study will be applied in practice and is expected to improve pesticide application.

We studied different models using canopy point clouds and leaf areas. However, the proposed model cannot be directly applied to other cultivars with higher/lower canopy densities or to different apple tree planting patterns. Therefore, new models need to be created before using this method for other cultivars. In this study the LiDAR instrument moved along the LiDAR detection mobile platform, which can avoid the influence of uneven ground conditions. The influence of topography will be considered in a future

larger-scale experiment that will investigate the orchard. Uneven ground may cause deviations in the angular orientation of the LiDAR sensor during scanning. Equipment such as an inertial measurement unit (IMU) or a global navigation satellite system (GNSS) device must be used with a LiDAR detection system for canopy leaf area detection models.

## 5. Conclusions

In this paper, leaf area detection models for the thick canopy in an apple orchard were studied, a LiDAR detection mobile test platform and a three-dimensional test platform for evaluating the canopy leaf area of orchard trees were built, and the LiDAR point cloud data and leaf areas in different areas of the orchard tree canopy were obtained. The residual method was used to remove outliers from the data, eliminate the influence of dense branches and leaves in the canopy, and obtain experimental data. Polynomial regression, the BP neural network, and the PLSR algorithms were used for leaf area modelling and regression analysis. Comparing the results for $R^2$ from the obtained models and their ability to predict data revealed that the BP neural network algorithm was better than the other two algorithms. With the BP neural network algorithm, the $R^2$ (86.1%) was greater than those obtained using the Fourier function (79.43%) and the PLSR algorithm (78.46%). The prediction accuracy of the model obtained with the BP neural network regression algorithm for the validation set data (73.6%) was much higher than those of the models obtained with polynomial regression and the PLSR algorithm for the measured and predicted values of the validation set data. Comparing the use of the PLSR and polynomial regression to obtain leaf area models revealed that the PLSR algorithm is a simple and intuitive means to obtain a model and has a strong predictive ability. Both the BP neural network and the PLSR algorithm can be considered for applications of variable spraying in orchards. The selection of the leaf area detection model needs to be combined with considerations of the difficulty of system control and programming and the accuracy required for spraying. In the operation process, if the accuracy requirements are not high and the system and operation accuracy are not particularly high, then selection of the regression model obtained using the partial PLSR algorithm is recommended. In contrast, the BP neural network regression model is recommended for applications that require a higher implementation accuracy. The BP method of establishing a model is universal and can provide a reference for the establishment of other orchard leaf area models. At the same time, combined with environmental protection and intelligent operation, this model can serve as a precise target variable spraying system, reduce pesticide application and residues, and achieve environmental protection and the control of plant protection against diseases. This model can form the basis of an intelligent system that can be used to control variables and achieve intelligent spraying of targets.

This paper describes a leaf area detection model, using a two-dimensional spatial function of the thick canopy of an apple orchard. In subsequent studies, the influence of canopy thickness needs to be considered. The study of leaf area detection models in three-dimensional space is an important research topic for optimizing leaf area detection models and improving the accuracy of model predictions. This article researched the feasibility of LiDAR point cloud data for a thick canopy, and the influence of LiDAR moving speed on the leaf area detection model can be researched in a further study.

**Author Contributions:** Conceptualization, C.Z. (Chunjiang Zhao) and C.G.; Methodology, C.G., C.Z. (Changyuan Zhai), W.Z., S.Y. and C.Z. (Chunjiang Zhao); Validation, H.D.; Formal analyses, C.Z. (Chunjiang Zhao), W.Z. and S.Y.; Investigation, C.G. and H.D.; Resources, C.Z. (Chunjiang Zhao) and C.Z. (Changyuan Zhai); Data curation, C.G.; Writing—original draft, C.G.; Writing—review and editing, C.Z. (Chunjiang Zhao), C.Z. (Changyuan Zhai), W.Z., S.Y. and H.D.; Funding acquisition, C.Z. (Chunjiang Zhao) and C.Z. (Changyuan Zhai); Supervision, C.Z. (Chunjiang Zhao) and C.Z. (Changyuan Zhai). All authors have read and agreed to the published version of the manuscript.

**Funding:** Support was provided by (1) the Special Key Project of Chongqing Technology Innovation and Application Development (Grant number: cstc2019jscx-gksbX0089); (2) the Natural Science Foundation of China (Grant number: 31971775); and (3) the Outstanding Scientist Program of Beijing Academy of Agriculture and Forestry Sciences (Grant number: jkzx202212).

**Institutional Review Board Statement:** Not applicable.

**Informed Consent Statement:** Not applicable.

**Data Availability Statement:** The data presented in this study are available on request from the corresponding author.

**Conflicts of Interest:** The authors declare no conflict of interest.

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
