# Peer review of "Innovative Leaf Area Detection Models for Orchard Tree Thick Canopy Based on LiDAR Point Cloud Data"

_agriculture, doi:10.3390/agriculture12081241_

Round 1

Reviewer 1 Report

The article is well written, it provides descriptive and detailed information about the scientific methods and analysis. The results are clear, quite interesting and may be considered novel for the case of spraying thick canopies of trees like apples. I only have some few notifications on the text:

The R2 of the model at site one was lower than that at site two; the main cause was the lower leaf foliage density of the trees with less overlap at site two

Instead of R2 I suggest to use the term “coefficient of determination (R2)”

 The grid area of the grid frame was 1.5 m high and 4 m wide

Instead of “area” use the term “dimensions”

 the width of the vertical and horizontal grid was 0.1 m

I suggest to use the phrase : “the grid mesh was 0.1m “

 The leaf area in each size grade, the weighted average value, and the leaf area in each size grade were calculated

The phrase “The leaf area in each size grade” is doubled. Please delete one.

 The outliers removal steps were calculate the parameters of the outlier removal model, calculate the data residuals, judge the outlier and remove the outliers.

Please revise the above sentence and make it more clear .

 In this paper, we presented describes our study

Please correct the phrase

Author Response

Article Title: “Innovative Leaf Area Detection Models for Orchard Tree Thick Canopy Based on LiDAR Point Cloud Data”

Dear Reviewer,

On behalf of all the authors, we sincerely appreciate your valuable comments on the manuscript. Your comments not only provide constructive suggestions on improving the quality of the manuscript but also lead us to consider our approaches and the design of the systems in detail. Our future research will benefit from these comments, as well.

Best regards,

Chenchen Gu, Chunjiang Zhao, Wei Zou, Shuo Yang, Hanjie Dou and Changyuan Zhai

1: “The R2 of the model at site one was lower than that at site two; the main cause was the lower leaf foliage density of the trees with less overlap at site two” Instead of R2 I suggest to use the term “coefficient of determination (R2)”

Author response: Thank you for the suggestion. We change R2 in the manuscript to “coefficient of determination (R2)”. (Yellow highlight in L83)

2: “The grid area of the grid frame was 1.5 m high and 4 m wide” Instead of “area” use the term “dimensions”

Author response: We change “The grid area” in the manuscript to “the grid dimensions”. (Yellow highlight in L135)

3: “the width of the vertical and horizontal grid was 0.1 m” I suggest to use the phrase: “the grid mesh was 0.1m”

Author response: We change “the width of the vertical and horizontal grid was 0.1 m” in the manuscript to “the grid mesh was 0.1m”. (Yellow highlight in L135-136)

4: “The leaf area in each size grade, the weighted average value, and the leaf area in each size grade were calculated” The phrase “The leaf area in each size grade” is doubled. Please delete one.

Author response: The sentence is simplified to “The leaf area of each stage was measured and weighted to average”. (Yellow highlight in L161-162)

5: “The outliers removal steps were calculate the parameters of the outlier removal model, calculate the data residuals, judge the outlier and remove.” Please revise the above sentence and make it more clear.

Author response: The sentence was revised. (Yellow highlight in L226-228)

6: “In this paper, we presented describes our study” Please correct the phrase

Author response: The phrase and the sentence were both revised. (Yellow highlight in L579-582)

Reviewer 2 Report

The manuscript by Gu et al. investigates the using of LiDAR for prediction of leaves area of apple trees. This work seems to be interesting for agriculture cultivation. However, I have some questions.

(1) How were measured the real leaves area of apple trees for LiDAR validation?

(2) How many plants were used in investigation?

(3) What is advantage of LiDAR using compared to RGB and others methods?

(4) The captions of Figures 1 and 2 should be expanded.

(5) The regression coefficients and RMSE should be added in Figure 10 for each regression curve.

(6) The regression coefficient and RMSE should be added in Figure 13.

(7) The relationship (scatter plot) between real and predicted values should be added in section 3.6 after Figure 15.

(8) Please, to discuss relation between leaf area of trees (plants) and pesticide spraying.

Author Response

Article Title: “Innovative Leaf Area Detection Models for Orchard Tree Thick Canopy Based on LiDAR Point Cloud Data”

Dear Reviewer,

On behalf of all the authors, we sincerely appreciate your valuable comments on the manuscript. Your comments not only provide constructive suggestions on improving the quality of the manuscript but also lead us to consider our approaches and the design of the systems in detail. Our future research will benefit from these comments, as well.

Best regards,

Chenchen Gu, Chunjiang Zhao, Wei Zou, Shuo Yang, Hanjie Dou and Changyuan Zhai

1: How were measured the real leaves area of apple trees for LiDAR validation?

Author response: The real leaves area of apple trees was calculated through three steps. Firstly, the leaves in the statistical area are divided into three grades: large, medium and small. Secondly, count the number of leaves in each grade. Last step, the calculated average leaf area of each grade was multiplied by the number of leaves in that grade. (Yellow highlight in L173-176).

In the process of “Model analysis of the relationship between LiDAR and LA”, 70% data was used for model establish and the rest 30% data was used for model validation. (Yellow highlight in L258-261)

2: How many plants were used in investigation?

Author response: We used one Fuji apple tree in the investigation, the tree was 5 years old, the canopy density was approximately average for the orchard. 240 groups data was obtained (Yellow highlight in L215-216), and the number data is enough in the process of data analyzing.

3: What is advantage of LiDAR using compared to RGB and others methods?

Author response: Thank you for the question. Both sensors can be used for canopy density detection. However, RGB cameras can be easily influenced by sunlight, and the process of image processing is complex. The LiDAR sensor has good stability and is rarely influenced by weather conditions. the point cloud data can be calculated quickly. (Yellow highlight in L39-41, L67-69).

4: The captions of Figures 1 and 2 should be expanded.

Author response: Thank you for the suggestion. The captions of Figures 1 and 2 were both expanded. (Yellow highlight in L131, L144-145)

5: The regression coefficients and RMSE should be added in Figure 10 for each regression curve.

Author response: Thank you for the suggestion. The regression coefficient and RMSE were add in Figure 10, and the description was added. (Yellow highlight in L394)

6: The regression coefficient and RMSE should be added in Figure 13.

Author response: Thank you for the suggestion. The regression coefficient and RMSE were add in Figure 13, and the description was added. (Yellow highlight in L430, L436-437)

7: The relationship (scatter plot) between real and predicted values should be added in section 3.6 after Figure 15.

Author response: Thank you for the suggestion. The relationship (scatter plot) between real and predicted values was added in Figure 16. (Yellow highlight in L467-475)

8: Please, to discuss relation between leaf area of trees (plants) and pesticide spraying.

Author response: Thank you for the suggestion. Leaf area is the base of spray deposition density calculate in the NY/T 992-2006 standard (China). The spray volume is required to achieve a spray deposition density of 25 droplets/cm2 for a low-volume spray and 70 droplets/cm2 for typical fungicides. Research on canopy leaf area detection models can provide a useful basis for deter-mining the spray volume. (Yellow highlight in L62-66)

Round 2

Reviewer 2 Report

Authors corrected manuscript in accordance with my comments. I have not other remarks.